# Evaluating the Growth Performance of Nile and Red Tilapia and Its Influence on Morphological Growth and Yield of Intercropped Wheat and Sugar Beet Under a Biosaline Integrated Aquaculture–Agriculture System

**DOI:** 10.3390/plants14091346

**Published:** 2025-04-29

**Authors:** Khaled Madkour, Fahad Kimera, Muziri Mugwanya, Rafat A. Eissa, Sameh Nasr-Eldahan, Kholoud Aref, Walaa Ahmed, Eman Farouk, Mahmoud A. O. Dawood, Yasmine Abdelmaksoud, Mohamed F. Abdelkader, Hani Sewilam

**Affiliations:** 1Center for Applied Research on the Environment and Sustainability (CARES), School of Science and Engineering, The American University in Cairo, AUC Avenue, P.O. Box 74, New Cairo 11835, Egypt; khaled_hafez@aucegypt.edu (K.M.); muziri@aucegypt.edu (M.M.); raafat.abdulmajeed@aucegypt.edu (R.A.E.); sameh.eldahan@aucegypt.edu (S.N.-E.); kholoudaref@aucegypt.edu (K.A.); walaa.ahmed@aucegypt.edu (W.A.); eman.farouk@aucegypt.edu (E.F.); mahmoud-dawood@aucegypt.edu (M.A.O.D.); ymaksoud@aucegypt.edu (Y.A.); 2Animal Production Department, Faculty of Agriculture, Kafrelsheikh University, Kafr El-Sheikh 33516, Egypt; 3Electrical Engineering Department, Faculty of Engineering, Port Said University, Port Said P.O. Box 42523, Egypt; mdfarouk@eng.psu.edu.eg; 4UNESCO Chair in Hydrological Changes and Water Resources Management, RWTH Aachen University, 52056 Aachen, Germany

**Keywords:** Nile tilapia, red tilapia, aquaculture–agriculture, sugar beet, wheat, salinity, organic

## Abstract

Integrated aquaculture–agriculture systems (IAASs) offer a sustainable approach to mitigating soil salinity by utilizing aquaculture effluents for irrigation. This study evaluates the growth performance of Nile tilapia (*Oreochromis niloticus*) and red tilapia (*Oreochromis* spp.) under varying salinity conditions and investigates their effluents on intercropped wheat and sugar beet. A field experiment was conducted using a randomized block design with seven treatments: control (chemical fertilizers dissolved in freshwater) and brackish water effluents from Nile tilapia and red tilapia at salinities of 5 ppt and 10 ppt as monocultures or mixed polycultures. Fish growth parameters were assessed, while wheat and sugar beet morphological and yield traits were monitored. Statistical analyses, including correlation and principal component analysis, were performed. Red tilapia outperformed Nile tilapia at 10 ppt salinity, achieving the highest final weight (174.52 ± 0.01 g/fish) and weight gain (165.78 ± 0.01 g/fish), while the mixed polyculture at 10 ppt exhibited optimal feed conversion (FCR: 1.32 ± 0.01). Wheat growth and yield traits (plant height, stalk diameter, and panicle weight) declined significantly under salinity stress, with 10 ppt treatments reducing plant height by ~57% compared to the control. Conversely, sugar beet demonstrated resilience, with total soluble solids (TSS) increasing by 20–30% under salinity. The mixed effluent partially mitigated salinity effects on wheat at 5 ppt but not at 10 ppt. This study highlights the potential of IAAS in saline environments, demonstrating red tilapia’s adaptability and sugar beet’s resilience to salinity stress. In contrast, wheat suffered significant reductions in growth and yield.

## 1. Introduction

Global food security is being increasingly threatened due to climate change and population growth. Soil degradation and salinization, exacerbated by unsustainable agricultural practices and changing climate conditions, further reduce agricultural productivity. This challenge is particularly significant for key staple crops such as wheat and sugar beet, which are essential to global food supply chains. Addressing the impacts of soil salinity on crop yield and exploring sustainable agricultural techniques are critical for ensuring long-term food security [1,2,3].

Soil salinity is one of the major abiotic stresses negatively affecting global agricultural production. High salinity levels disrupt plant growth by inducing osmotic stress and ion toxicity, leading to oxidative damage in cellular components such as proteins, enzymes, and DNA. These disruptions interfere with metabolic processes, ultimately reducing plant growth and crop yield [4,5]. Additionally, salinity alters soil structure by causing clay particle aggregation, increasing soil density, and reducing water drainage capacity [6,7].

Currently, one-third of irrigated agricultural lands and one-fifth of cultivated lands worldwide are affected by salinity. Soil salinization rates are projected to increase significantly by 2050, driven primarily by excessive chemical fertilizer application and groundwater use to support growing populations [8,9]. Approximately 600 million people living in coastal areas are at risk due to the progressive salinization of agricultural lands [10]. Globally, 954 million hectares (Mha) of land are impacted by salinization across 120 countries, resulting in an estimated 7–8% reduction in agricultural productivity. In Africa alone, 80.4 Mha of agricultural land is affected [8,11,12].

To mitigate these challenges, innovative and resource-efficient solutions such as biosaline integrated aquaculture–agriculture systems (IAASs) represent a promising strategy to enhance food production sustainably. IAA is defined by the concurrent or sequential linkage of aquaculture and agricultural activities, creating synergies where the output or waste from one component becomes valuable input for another. This integration promotes on-farm waste recycling, improves resource use efficiency (land, water, and nutrients), increases overall productivity and profitability, reduces reliance on external inputs like chemical fertilizers, and minimizes environmental pollution [13,14]. A key principle is the multiple uses of water: water first supports fish culture, and the resulting nutrient-rich effluent is then reused for crop irrigation, embodying the “more crop per drop” concept and enhancing water productivity. Common IAA practices include using fish effluent to fertilize crops like vegetables, rice, or fodder or using agricultural by-products such as fish feed [15,16,17].

Salinity affects vast areas of cultivated land globally [18,19], necessitating the adaptation of agricultural practices [1]. Biosaline agriculture focuses on utilizing saline land and water resources productively, often employing salt-tolerant crops (halophytes or tolerant varieties of conventional crops) [20,21,22]. Integrating IAA principles within this context (Biosaline IAA) offers a powerful approach to valorizing these marginal resources [23]. In Biosaline IAA, saline water unsuitable for conventional crops can be used for culturing salt-tolerant fish species [24]. The nutrient-laden saline effluent from the fish culture is then channeled to irrigate and fertilize salt-tolerant crops, effectively recycling both water and nutrients while mitigating the environmental risks associated with effluent discharge [14]. While Biosaline IAA holds significant potential, scaling up these systems requires addressing challenges such as managing potential salt accumulation from effluent irrigation, high operational costs, and the need for specific technical expertise [25,26].

Tilapia are frequently used in IAA systems due to their adaptability [16]. However, different tilapia types exhibit varying traits crucial for biosaline systems. Nile tilapia (*Oreochromis niloticus*), the most widely farmed globally, is known for rapid growth but has limited salinity tolerance, with growth typically inhibited above ~15 ppt [27,28,29,30]. In contrast, red tilapia (*Oreochromis* spp.), often hybrids, generally exhibit superior tolerance to higher salinity levels, making them potentially more resilient in biosaline aquaculture, although their growth rates can be variable and sometimes slower than Nile tilapia. Furthermore, the physiological response to salinity might differ between these tilapia types, potentially affecting the nutrient composition of their effluent. Choosing the optimal tilapia type for a biosaline IAA system involves a trade-off between growth potential and salinity resilience, highlighting the need for direct comparative studies within the integrated system context [31,32].

Similarly, crop selection is critical for successful Biosaline IAA. This study focuses on wheat (*Triticum aestivum*) and sugar beet (*Beta vulgaris*). Sugar beet is recognized for its relatively high salt tolerance [33]. It ranks second after sugarcane in global sugar production, contributing 20–40% of total sugar output. Additionally, sugar beet serves as a raw material for biofertilizers, bioethanol, and biodegradable polymers, making it a suitable candidate for biosaline agriculture [33,34,35,36,37]. Wheat, a global staple essential for food security, is considerably more sensitive to salinity, with yields declining significantly at moderate levels. Wheat, in contrast, is highly sensitive to salinity stress. As the primary staple food for 35% of the global population, wheat production plays a crucial role in food security. Global wheat cultivation spans 217 Mha, with an annual production of 752 million metric tons [38]. However, salinity levels above 4.8–6.4 ppt significantly reduce wheat yields by disrupting physiological and biochemical processes [39,40]. Including both a tolerant (sugar beet) and a moderately tolerant/sensitive (wheat) crop allows for a comprehensive assessment of how effluent from different tilapia types, cultured under salinity, impacts crops with varying physiological responses to salt stress. Evaluating not just the final yield but also morphological growth parameters provides deeper insights into plant responses to the effluent under these conditions.

Despite the growing interest in IAA and biosaline agriculture, a significant research gap exists. While studies have examined components in isolation (e.g., tilapia salinity tolerance, crop response to wastewater, and general IAA benefits), there is a lack of comprehensive, comparative research evaluating the performance of different tilapia species (Nile vs. red) within a biosaline IAA system while simultaneously assessing the specific influence of their respective effluents on the morphological growth and yield of inter-cropped salt-tolerant and salt-sensitive crops like sugar beet and wheat. Understanding these specific interactions is crucial for optimizing species selection and management practices in these integrated systems. Limited studies have investigated the effects of saline water on both fish and crops within these systems. Previous research has mainly focused on integrating aquaculture with rice, livestock, and specific crops such as lettuce in aquaponic systems [41,42].

This study aimed to compare the performance of Nile tilapia and red tilapia reared under defined biosaline conditions within an integrated system and evaluate the influence of effluent water derived from Nile tilapia versus red tilapia cultures on the morphological growth parameters of inter-cropped wheat and sugar beet under biosaline conditions. This research will provide insights into optimizing resource utilization and enhancing food production resilience in challenging saline environments through integrated aquaculture–agriculture practices.

## 2. Results

### 2.1. Fish Growth

Fish growth performance varied significantly across treatments and fish types (Table 1). Among treatments, red tilapia at 10 ppt salinity exhibited the highest final weight (FW) of 174.52 ± 0.01 g/fish and weight gain (WG) of 165.78 ± 0.01 g/fish, significantly surpassing all other treatments (*p* ≤ 0.05). The mixed treatment (Mix) at 10 ppt followed closely, achieving an FW of 171.57 ± 0.01 g/fish and WG of 162.81 ± 0.01 g/fish, highlighting strong growth potential under these conditions. Feed intake (FI) was highest in red tilapia at 5 ppt (232.05 ± 0.01 g/fish); however, this did not correspond to the greatest weight gain, suggesting that increased feed consumption alone did not directly translate to improved growth. The most efficient feed conversion was observed in the Mix treatment at 10 ppt, which had the lowest feed conversion ratio (FCR) of 1.32 ± 0.01, significantly lower than other treatments (*p* ≤ 0.05), indicating optimal feed utilization. The specific growth rate (SGR) was highest in the Mix treatment at 10 ppt (1.41 ab ± 0.01%/day), with red tilapia at the same salinity achieving a comparable rate of 1.42 a ± 0.01%/day. Survival rates (SRs) were also highest in the Nile tilapia and Mix treatments at 10 ppt, with both groups achieving a survival rate of 94%, suggesting that higher salinity conditions were not detrimental to these fish.

### 2.2. Growth Parameters of Wheat

All treatments significantly (*p* ≤ 0.05) altered plant growth traits, plant height, stalk diameter, and leaf area across the growth period (Figure 1). These traits generally decreased with increasing salinity, with 10 ppt treatments showing the most substantial reductions. Plant height was highest in the control group at all DAS points, decreasing values from ~90 cm (30 DAS) to ~70 cm (90 DAS). The 10 ppt treatments showed the lowest plant heights, particularly at 30 DAS (~20 cm), with values increasing slightly by 90 DAS (~30 cm). Stalk diameter followed a similar pattern, with control plants maintaining the thickest stalks (~3.5 cm at 30 DAS decreasing to ~3.0 cm at 90 DAS). The 10 ppt treatments had the thinnest stalks (~2.0 cm at 30 DAS). Leaf area was consistently highest in the control group across all DAS points, with the greatest differences observed at 30 DAS (~35 cm^2^ vs. ~10 cm^2^ in 10 ppt treatments). By 90 DAS, leaf area differences between treatments became less pronounced. The Mix treatment at 5 ppt showed intermediate values between control and single effluent treatments for most metrics, particularly at 30 DAS. Notably, the 10 ppt treatments underperformed across all metrics, indicating that higher salinity levels have compounding negative effects on plant development regardless of effluent type.

### 2.3. Yield Traits and Responses in Wheat

All salinity treatments (5 ppt and 10 ppt) significantly (*p* ≤ 0.05) altered the wheat plants’ growth parameters compared to the control treatment (Figure 2). Panicle length (Figure 2A), width (Figure 2B), and weight (Figure 2C) all decreased with increasing salinity, with the greatest reductions observed in the NT and RT treatments. At 10 ppt, the panicle length in NT and RT treatments was approximately 20–30% lower than the control, while the panicle weight showed even more pronounced declines, with NT and RT treatments at 10 ppt exhibiting values less than half of the control. The Mix treatment generally maintained intermediate values between the control and the more vulnerable NT/RT treatments at 5 ppt but converged with NT/RT at 10 ppt.

Root length (Figure 2D) and width (Figure 2E) followed similar patterns, with significant reductions under salinity stress. The control treatment maintained the highest root length and width across all salinity levels, while NT and RT treatments showed the most substantial decreases, particularly at 10 ppt, where root length was reduced by approximately 30–40% compared to the control. The Mix treatment again showed intermediate values at 5 ppt but aligned more closely with NT/RT at 10 ppt. Seed-related metrics, including 100-seed weight (Figure 2F), shoot weight (Figure 2G), and number of panicles per hill (Figure 2I), exhibited significant declines under salinity stress, with NT and RT treatments being the most affected. Notably, the number of panicles per five hills (Hs) showed a different pattern, with the Mix treatment maintaining values closer to the control at 5 ppt but still showing a significant reduction at 10 ppt.

These findings indicate that effluent type interacts with salinity stress in a trait-specific manner, with the Mix treatment providing partial mitigation at lower salinities but failing to confer protection at higher levels. The consistent vulnerability of NT and RT treatments across most parameters suggests these effluents may not adequately support plant growth under salinity stress, while the control treatment consistently outperformed all effluent treatments across nearly all measured parameters.

### 2.4. Morphological Parameters of Sugar Beet at Different Stages

Plant height exhibited the highest mean values in the control treatment across most days after sowing (DAS) at 75, 105, 135, and 165. At 30 DAS, the red tilapia (RT) group showed the tallest plants (28.5 ± 0.75 cm), significantly higher than all other treatments. By 45 DAS, the red tilapia effluent at 5 ppt demonstrated the greatest height (40.42 ± 0.54 cm). At 75 DAS, the control treatment regained the highest value (47.67 ± 0.75 cm), with red tilapia at 5 ppt (44.50 ± 0.15 cm) and Mix at 5 ppt (46.50 ± 0.86 cm) showing competitive growth. The 10 ppt treatments generally showed intermediate values, with red tilapia at 10 ppt reaching 49.62 ± 0.85 cm at 165 DAS, comparable to the control’s 53.42 ± 0.74 cm. This is well presented, as shown in Table 2.

The number of leaves increased in the salinity treatment (Table 3), and the 10 ppt treatment consistently produced the highest number of leaves across all treatments except at 30 DAS. At 30 DAS, the Mix treatment had 8.42 ± 0.31 leaves, significantly more than other treatments. By 45 DAS, red tilapia at 5 ppt showed 10.00 ± 0.48 leaves, matching the control’s 10.33 ± 0.48 leaves. At 75 DAS, the control maintained the lead with 15.17 ± 0.53 leaves, while red tilapia at 5 ppt (16.75 ± 0.65 leaves) and Mix at 5 ppt (15.75 ± 0.71 leaves) showed strong performance. The 10 ppt treatments demonstrated gradual improvement, with Nile tilapia at 10 ppt reaching 24.58 ± 0.82 leaves at 135 DAS and 38.33 ± 0.59 leaves at 165 DAS, approaching the control’s 36.75 ± 0.71 leaves.

### 2.5. Yield Traits of Sugar Beet

All salinity treatments (5 ppt and 10 ppt) significantly affected plant yield traits, including leaf dry weight, leaf fresh weight, root weight, root length, root diameter, and total soluble solids (TSS) compared to the control treatment (Figure 3). Leaf dry weight and fresh weight showed substantial reductions under salinity stress, with the greatest declines observed in the NT and RT treatments, particularly at 10 ppt, where these treatments exhibited approximately 40–50% and 60–70% reductions, respectively, compared to the control. Root traits followed similar patterns, with root weight, length, and diameter all decreasing with increasing salinity, though the Mix treatment often maintained intermediate values between the control and the more vulnerable NT/RT treatments at 5 ppt. Notably, the TSS displayed an opposing trend, increasing significantly under salinity stress across all treatments, with the NT and RT treatments showing the highest values. The Mix treatment generally bridged the gap between the control and the NT/RT groups at moderate salinity but converged with NT/RT at 10 ppt, suggesting a threshold effect at higher salinity levels. These findings indicate that effluent type interacts with salinity stress in a trait-specific manner, with the Mix treatment providing partial mitigation at lower salinities but failing to confer protection at higher levels, while TSS accumulation highlights a divergent physiological response to stress.

### 2.6. Principal Component Analysis for Both Wheat and Sugar Beet

Principal component analysis (PCA) of integrated growth and yield traits reveals distinct clustering patterns across treatment conditions (Figure 4). Figure 4A shows the separation of data points along PC1 (37.3%) and PC2 (19%), with control treatments forming tighter clusters compared to salinity-stressed groups. The orientation of trait vectors indicates shifts in associations: under control conditions, growth-related parameters (LDW, LFW, and R_Weight) aligned positively with PC2, while under salinity stress, these traits rotated toward negative PC1 values, suggesting altered physiological priorities. Micronutrients (Mn, Fe, Cu, and Zn) formed a separate cluster at positive PC2 values across all treatments, indicating their relatively stable association despite salinity changes. Figure 4B further illustrates how treatment conditions modulate trait expression, with the 10ppt_Mix treatment showing the most pronounced divergence from control conditions at the positive end of PC1 (39.6%). The dispersion of data points demonstrates a clear separation between control treatments and those exposed to 5 ppt and 10 ppt salinity, with the latter showing greater spread along both principal components. These patterns demonstrate that salinity stress reshapes the multivariate relationships among physiological traits, with the Mix treatment at higher salinity exhibiting a unique metabolic profile distinct from other treatments. The PCA further highlights how effluent type modulates trait expression under salinity stress, with the Mix treatment showing the most pronounced divergence from control conditions at 10 ppt.

These patterns demonstrate that salinity stress reshapes the multivariate relationships among physiological traits, with the Mix treatment at higher salinity exhibiting a unique metabolic profile distinct from the others.

### 2.7. Correlation Analysis Between Wheat Traits Under Different Treatments

The correlation matrix provides a comprehensive overview of relationships between growth and yield parameters, nutrient concentrations, and kernel nutrients in wheat plants across different treatments. Strong positive correlations were observed among key growth indicators, with plant height (PH) showing significant relationships with leaf area (LA) and root length (RL), indicating coordinated growth patterns throughout plant development. Yield parameters exhibited distinct correlation patterns, with panicle weight (P_Weight) demonstrating strong positive relationships with the number of panicles per five hills (NP/5H) and shoot weight (ShW). These correlations indicate that resource allocation to reproductive structures is closely linked to overall plant biomass production as shown in Figure 5.

Kernel nutrient concentrations showed complex interactions with the growth and yield parameters. Macronutrients such as nitrogen (N) and phosphorus (P) displayed positive correlations with yield traits (P_Weight and NP/5H). Potassium (K) demonstrated positive correlations with the growth parameters (PH: 0.20 and LA: 0.18). Similarly, micronutrients like iron (Fe) and zinc (Zn) showed significant relationships with the stress-related parameters, including malondialdehyde (MDA) and proline content, while sodium (Na) exhibited negative correlations with most growth traits, consistent with their respective roles in plant physiology under stress conditions.

## 3. Discussion

This study aimed to evaluate the performance of Nile and red tilapia reared in brackish water with a salinity level of 5 and 10 ppt and sugar beet intercropped with wheat plants under different irrigation conditions using saline fish effluents from Nile and red tilapia and mixed polyculture effluent.

Due to salinity variations, significant changes were observed in the fish growth metrics of Nile tilapia, including final body weight (FW), weight gain (WG), feed intake (FI), and specific growth rates (SGRs), which were consistent. As the salinity concentration increased, the growth metrics of Nile tilapia decreased, which is consistent with [43]. This aligns with known physiological limitations; while Nile tilapia (*Oreochromis niloticus*) can adapt to moderate salinities, its growth performance is typically inhibited at salinities above approximately 15 ppt due to the increased metabolic cost of osmoregulation to maintain a stable internal balance of salts and water, which is physiologically demanding for fish, particularly when the external salinity deviates significantly from their internal osmotic concentration [44,45,46,47], This process requires the active transport of ions across membranes, primarily in the gills, which consumes substantial metabolic energy in the form of ATP [45,47]. In addition, typically exhibit poorer (higher) feed conversion ratios (FCRs), meaning more feed is required to produce a unit of weight gain [48]. Feed intake (FI) may also decrease at these higher salinities, further contributing to reduced growth [49]. Conversely, red tilapia growth metrics were positively correlated with the salinity treatment used in this study, as shown in Table 1, which was also mentioned by [44,50] since moderate salinity treatments do not affect the performance of red tilapia until an optimal level. This superior performance under saline conditions is characteristic of many red tilapia strains, which are often hybrids incorporating genetics from more salt-tolerant species like *Oreochromis mossambicus* [32,51]. This genetic background likely confers a more efficient osmoregulatory capacity, allowing them to maintain growth and physiological function better than Nile tilapia at elevated salinities, often thriving in conditions well above 15 ppt [51]. The observation that moderate salinity did not negatively affect but potentially enhanced red tilapia performance up to the tested levels is consistent with findings suggesting that optimal ranges for certain strains can fall within brackish conditions [32,51].

Regarding wheat growth and yield performance, our findings indicate that salinity significantly affected wheat growth metrics, including plant height, stalk diameter, and leaf area (Figure 1A–C), with the most pronounced reductions noted in the highest salinity treatment (10 ppt). The control group consistently displayed the highest values for all measured parameters, highlighting the detrimental effects of salinity on plant development. This aligns with prior research suggesting that wheat is moderately sensitive to salinity since increased salinity levels lead to diminished growth and yield [52]. Higher salinity levels can hinder plant growth due to osmotic stress, making it difficult for the plant roots to absorb water, leading to physiological drought and reduced pressure necessary for cell expansion and growth [25]. Excessive uptake and accumulation of specific ions, particularly sodium (Na^+^) and chloride (Cl^−^), result in ion toxicity, disrupting essential enzymatic activities, damaging cellular structures, and interfering with metabolic processes [53]. This toxicity also disrupts ion homeostasis, particularly the crucial balance between potassium (K^+^) and Na^+^, impacting nutrient uptake and transport. Salinity can also alter soil physical and chemical properties, potentially reducing the availability of essential nutrients. These primary stresses often lead to secondary oxidative stress through the overproduction of reactive oxygen species (ROS), further damaging cellular components. All yield traits exhibited a drastic decrease due to salinity treatment, but a variation in fish effluent impacted a slight change in panicle width (Figure 2B); this drastic decrease was discussed by [54], who demonstrated that the translocation of water-soluble ions, i.e., salt ions, was limited to the xylem, which tends to accumulate in the root. In addition, a higher effect was observed on the phloem; therefore, the transport of photosynthetic materials to the young roots and plant organs decreased.

The varied response on the level of fish effluent may be due to the change in water quality [55]. Notably, the Nile tilapia effluent maintained a decrease in EC and the level of K^+^ and Na^+^ in the soil, which impacted the growth of plants (Figure 2). K^+^/Na^+^ has an osmotic stress potential on plants, highlighted in the interference of the uptake of other essential nutrients. Improving this ratio can help mitigate the effect of salinity on plant growth [56,57] compared with other effluents, suggesting that using fish effluent may reduce the harmful impact of saline water.

On the other hand, sugar beet is known to be more saline-tolerant compared to wheat. Plant height decreased due to exposure to salinity stress; however, this reduction was more dramatic under the 5 ppt Nile tilapia effluent (Table 2). Among sugar beet traits, the number of leaves and root characteristics and TSS showed higher mean values when compared with the control fertilizer (Table 3 and Figure 3C–E). Irrigating sugar beet with saline effluent, the roots often display a pronounced rise in total soluble solids (TSS) (Figure 3F) compared with the control effluent, mainly because the plants accumulate soluble sugars and other compatible solutes to maintain osmotic balance and mitigate ion toxicity. This adaptive mechanism not only helps the crop tolerate salt stress but also boosts sugar extraction efficiency by increasing the concentration of sugars within the beetroot. For instance, research has shown that sugar beet grown in a mildly saline environment can exhibit higher TSS levels without incurring significant yield penalties, reflecting the resilience and capacity for osmotic adjustment [58]. Therefore, sugar beet can grow under our salinity treatments, which was in line with [59], who demonstrated that sugar beet can enhance its growth rates and germination under low salt concentration, promoting root elongation and increased surface area, which allow it to absorb more water efficiently, highlighting that sugar beet is tolerant to salt stress [60,61,62].

In this study, the PCA analysis demonstrates that salinity stress drastically alters the multivariate relationships among wheat’s growth and yield traits (Figure 4). Under control conditions, the data points cluster tightly, indicating a uniform expression of growth parameters such as leaf dry weight, leaf fresh weight, and root weight. In contrast, under salinity stress, these traits shift significantly, as shown by their rotation toward negative PC1 values (Figure 4A), suggesting that wheat plants reallocate resources to cope with the osmotic and ionic challenges imposed by salt [63]. Furthermore, the consistent clustering of micronutrients (Mn, Fe, Cu, and Zn) at positive PC2 values implies that, despite salinity, the uptake and internal balance of these essential elements remain relatively stable, which aligns with previous findings that efficient micronutrient management contributes to improved salt tolerance in wheat [56]. Figure 4 shows that the 10 ppt Mix treatment diverges most from control conditions at the positive end of PC1 (39.6%). Data dispersion indicates a clear separation between control treatments and those exposed to 5 ppt and 10 ppt salinity, with higher salinity causing greater spread along both principal components. These patterns demonstrate that salinity stress reshapes the relationships among physiological traits, with the Mix treatment at higher salinity exhibiting a unique metabolic profile distinct from other treatments. Salinity stress alters fish metabolism and energy expenditure [45,64]. This can influence the rate and form of nutrient excretion. For instance, increased protein catabolism for energy might increase ammonia excretion, while reduced feed intake under severe stress could decrease overall waste output.

We investigated the performance of wheat plants intercropped with sugar beet under different salinity levels (5 and 10 ppt). The results reveal the significant impacts of salinity on wheat growth and yield parameters, consistent with previous research showing that salinity stress disrupts plant–water relations and ion homeostasis, leading to reduced biomass production and yield [65]. Plant height (PH), leaf area (LA), and root length (RL) showed substantial reductions under salinity stress, particularly at 10 ppt. This aligns with findings that high salinity inhibits cell division and elongation, thereby stunting overall plant growth [66]. Such an effect may be due to the influence of salinity on nutrient solubility and chemical speciation, such as the toxicity of ammonia is affected by salinity due to shifts in the equilibrium between unionized (NH_3_, more toxic) and ionized (NH_4_) forms [67]. Aquaculture wastewater is composed of several nutrients such as nitrogen, phosphorus, iron, and other elements in addition to organic matter, undecomposed feeds, feces, and dead fish tissues, which may have a significant impact on the health and soil nutritional status that eventually results in a paramount impact on crop growth. The strongest negative correlations were observed between salinity levels and these growth parameters, highlighting their sensitivity to osmotic stress. Panicle weight (P_Weight), number of panicles per five hills (NP/5H), and shoot weight (ShW) also exhibited significant declines under salinity stress. These yield components are particularly susceptible to salinity, as stress during reproductive stages can disrupt pollen fertility and grain filling [68]. Our correlation analysis reveals strong positive relationships between these yield parameters and growth metrics, suggesting that salinity’s impact on yield is closely linked to its effects on vegetative growth. Nutrient dynamics under salinity stress and kernel nutrient concentrations showed distinct responses to salinity stress. Macronutrients such as nitrogen (N) and phosphorus (P) displayed positive correlations with yield parameters, consistent with their essential roles in protein synthesis and energy transfer. However, salinity stress reduced the accumulation of these nutrients in wheat plants, likely due to impaired root function and reduced water uptake [54,65]. Micronutrients like iron (Fe) and zinc (Zn) displayed complex relationships with the growth and yield parameters (Figure 6). While these nutrients are critical for enzyme function and stress responses, their availability can be affected by soil salinity and pH changes [65]. Our results indicate that Fe and Zn concentrations in wheat kernels were negatively correlated with salinity levels, suggesting that salinity may limit their uptake or translocation to grains. Potassium (K) exhibited positive correlations with growth parameters, while sodium (Na) showed negative correlations with most growth traits. This pattern reflects the antagonistic relationship between these ions in plant physiology, where K^+^ is essential for enzyme activation and osmoregulation, while Na+ accumulation can be toxic at high concentrations.

Biosaline integrated agriculture-aquaculture systems, exemplified by the tilapia–wheat/sugar beet model, represent a significant opportunity to enhance food and feed production on marginal lands while improving resource use efficiency. As climate change intensifies water scarcity and soil salinization in many agricultural regions, the ability of these systems to utilize saline water resources productively will become increasingly valuable. They offer a pathway toward climate resilience for vulnerable farming communities by diversifying production and income streams.

However, realizing this potential requires overcoming the significant technical and management challenges associated with salinity. Success will depend on continued research to refine our understanding of the complex interactions involved, the development of adapted technologies and genotypes, and, crucially, effective knowledge transfer and capacity building supported by enabling policies. A holistic approach, considering the ecological (water quality, soil health, and biodiversity), economic (profitability and market access), and social (farmer livelihoods and equity) dimensions, is essential for the sustainable scaling and long-term contribution of biosaline IAA to regional and global food security, aligning with the Sustainable Development Goals related to zero hunger (SDG2), clean water and sanitation (SDG6), and responsible consumption and production (SDG12). The transition toward such integrated systems requires a paradigm shift from viewing aquaculture effluent as waste to recognizing it as a valuable resource, albeit one that requires careful management in saline contexts.

## 4. Materials and Methods

### 4.1. Site Description

The current field study was carried out in the winter growing season between September 2023 and April 2024 at the Center for Applied Research on the Environment and Sustainability (CARES), The American University in Cairo, New Cairo, Egypt (30°01′11.7″ N 31°29′59.8″ E). The climate data during the experimental period are presented in Table 4.

### 4.2. Experimental Design

The study investigated growing two different fish species, Nile tilapia (*Oreochromis niloticus*) and red tilapia (*Oreochromis* spp.), under two different brackish water salinities (5 ppt and 10 ppt). The effluents from aquaculture were used to irrigate two crops in an intercropping pattern, sugar beet (*Beta vulgaris* L.) (Novatelka KWS variety) and wheat (*Triticum aestivum* L.) (Misr 1 variety). The experiment followed a randomized completely block design of seven irrigation treatments, namely, control (chemical fertilizers dissolved in freshwater), Nile tilapia brackish water fish effluents at 5 ppt, red tilapia brackish water fish effluents at 5 ppt, a 1:1 polyculture of Nile and red tilapia under brackish water at 5 ppt, Nile tilapia brackish water fish effluents at 10 ppt, red tilapia brackish water fish effluents at 10 ppt, and a 1:1 polyculture of Nile and red tilapia under brackish water at 10 ppt (Figure 6). Aquaculture wastewater in all treatments was prepared by culturing 100 fish with an average weight of 8.5 g in a 1 m^3^ water tank mixed with sea salt to the desired salinity levels. The sugar beet and wheat seeds were acquired from the Agricultural Research Center (ARC) in Giza, Egypt.

### 4.3. Agronomical Procedures

The seeds of wheat and sugar beet were planted in an intercropping pattern by hand in rows where the space of inter- and intra-row was 15 cm and 50 cm, respectively. A fixed number of wheat seeds was sown in all treatments. The experiments were performed in isolated, separate plots, measuring 4 m × 3.5 m. Plants in all treatments were irrigated by an automated drip irrigation system as per the crop water requirements. Both crops were irrigated with effluent water from fish tanks in an open-loop system as per their corresponding treatments, and the same amount of water taken out for irrigation was compensated back to the fish tanks accordingly. Fish were acclimatized in their respective salinity treatments within three weeks after planting the crops, with a gradual increase in water salinity until the targeted levels. Irrigation was initially started after seedling emergence during the gradual increase in salinity levels of the fish tanks after the fish had been cultivated in the water tanks for one week. Insect pests and diseases were controlled according to the recommendations of the Egyptian Ministry of Agriculture. Similarly, the irrigation and fertilizer application of the wheat and sugar beet were applied according to the Egyptian Ministry of Agriculture guidelines. Thinning was performed on the sugar beet plants three weeks after sowing to uniformly keep one plant per hill.

### 4.4. Trait Measurement

A total of six plants within each replicate’s border were randomly tagged. During every stage of the data collection of sugar beet (30, 45, 75, 105, 135, and 165 days after sowing (DAS)), the plant height and number of leaves were determined. At each data collection phase of wheat (30, 60, and 90 days after sowing (DAS)), the plant height, stalk diameter, and leaf area were measured. Plant heights of both wheat and sugar beet were obtained by measuring with a meter rule from the plant’s crown to its terminal growing tip. The leaf number of sugar beet was acquired by counting healthy full-grown leaves per plant, and the averages were calculated. A digital vernier caliper was used to measure the diameter of each of the stalks of wheat at their mid-centers. The leaf area of wheat was evaluated according to the following equation:Leaf area = L × W × C
where L is the leaf length, W is the leaf width, and C is the constant (0.75).

For the sugar beet, after reaching maturity, six plants were chosen at random from each replicate. Each plant’s fresh leaf weight was determined using a high-precision digital scale (precise to 0.01 g). The average leaf weight for each treatment was determined. For the measurement of fresh forage yield, the leaves of six plants from each experimental plot were collected at the harvesting stage. The fresh weight of this forage was recorded. Subsequently, the collected leaves were dried in an oven at 70 °C until the weight remained constant. The root weight was measured using a high-precision digital scale (accurate to 0.01 g), and the average root weight was computed. The TSS (%) was obtained with a manual refractometer [69]). The root systems were harvested, removed, cleaned, and weighed in kilograms individually at a time [69]:$$\mathrm{Root}\;\mathrm{yield}\;(\mathrm{ton}\;{\mathrm{ha}}^{-1})=\frac{Root\;yield\;(Kg)\times 10,000}{1000\times Plot\;size}$$

### 4.5. Nutrient Composition of Forage Biomass

A microwave digestion system Speed Wave Entry DAP-60 K (Berghof, Germany), was used to break down the wheat grain samples in an acidic solution. A 300 mg wheat sample placed in a digestion vessel was treated with a 3 mL solution containing 65% nitric acid (HNO_3_) and 35% hydrogen peroxide (H_2_O_2_). Using a clean glass bar, the liquid was gently shaken and stirred for ten minutes. The sample was heated in the microwave while the vessel was sealed. Following cooling, an Agilent 4210 MP-AES (Agilent Technologies, Inc., Santa Clara, CA, USA), equipped with a OneNeb Series 2 nebulizer (Ingeniatrics Tecnologías S.L., Sevilla, Spain) and a double-pass cyclonic spray chamber (Agilent Technologies, Inc.), was utilized to analyze the nutritional composition of the resultant clear solution. A nitrogen supply was provided using an Agilent 4107 Nitrogen Generator (Agilent Technologies, Inc.). The selection of wavelengths was determined from the MP Expert software library (v1.6.0.9255) based on the required sensitivity.

### 4.6. Statistical Analysis

All the statistical analysis was performed using R (v4.4.3, R Core Team 2025) in RStudio (v2024.12.1). The data obtained were subjected to two-factor (salt concentration × fish effluent) mean differences and were compared by an LSD test using the agricolae R package (v1.3-7) [70]. Differences of *p* < 0.05 were considered. For visualizing the descriptive statistics of the traits, we used box and whisker plots and line charts. These visualizations were created using the ggplot2 package [71]. To understand how the traits were related to each other, we calculated the correlation coefficients. The correlation matrix and heatmap were generated using the ggpair function from the GGALLY and ggplot2 packages [72].

## 5. Conclusions

This study investigated the performance of Nile tilapia and red tilapia within a biosaline integrated aquaculture–agriculture system (IAAS), examining the effects of their effluent on intercropped wheat and sugar beet. Our findings demonstrate the distinct responses of both fish species and crops to salinity. Red tilapia showed better growth potential under the tested saline conditions compared to Nile tilapia, likely due to its superior ability to manage salt balance. Wheat growth and yield were significantly reduced by salinity, confirming its moderate sensitivity, while sugar beet exhibited greater tolerance, aligning with its known characteristics. Irrigation with fish effluent provided nutrients but also introduced salt stress, highlighting the need for careful management in biosaline IAASs. The nutrient contribution from fish effluent, particularly organic matter, which can improve soil structure, offers a potential way to lessen the negative impacts of saline water on crop growth when managed properly. While this study provides valuable insights, further research is needed to fully optimize these integrated systems for better efficiency and long-term sustainability. More detailed studies are needed to understand the exact nutrient mix (N, P, K, and micronutrients) and salt levels (especially the K^+^/Na^+^ ratio) in water from different tilapia types under various salinity levels. How fish stress affects the water quality also needs more investigation. Also, we need long-term experiments to see how using salty fish water affects soil health over time, looking at salt build-up, soil structure, and soil microbes. Understanding how organic matter in the effluent helps wash salt away is particularly important.

This study emphasizes that salinity levels differentially affect the growth of Nile and red tilapia, with red tilapia showing better adaptability to higher salinity. Wheat plants are adversely affected by increased salinity, while sugar beet exhibits notable tolerance. The use of saline fish effluents for irrigation presents both challenges and opportunities, depending on the crop’s salinity tolerance. These insights are valuable for developing integrated aquaculture–agriculture systems in regions with saline water resources. Finally, finding the best balance between the number of fish and the area of crops, selecting the most salt-tolerant crop varieties and fish strains, and evaluating the overall costs and benefits are crucial next steps. Addressing management complexity and supporting farmer adoption through training and policy is also vital.

## Figures and Tables

**Figure 1 plants-14-01346-f001:**
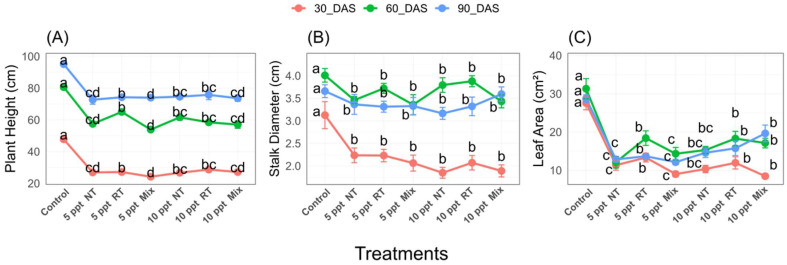
Effects of different treatments on wheat traits: (**A**) plant height, (**B**) stalk diameter, and (**C**) leaf area. Line chart displays the measured values of various traits across three main salinity treatments: 10 ppt, 5 ppt, and control, with seven irrigation effluent sub-treatments: NT (Nile tilapia), RT (red tilapia), and Mix (Nile and red tilapia). DAS refers to days after sowing. Different letters indicate significant differences between sub-treatments at the same stage (DAS) (*p* < 0.05).

**Figure 2 plants-14-01346-f002:**
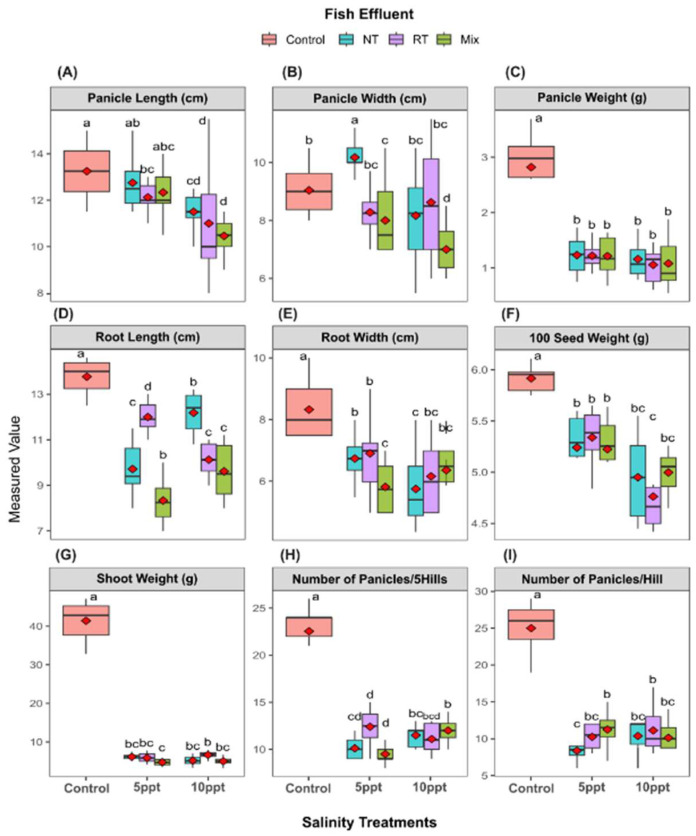
Effects of different treatments on wheat growth and yield traits. Boxplots display the measured values of various traits across three main salinity treatments: 10 ppt, 5 ppt, and control, with seven irrigation effluent sub-treatments: NT (Nile tilapia), RT (red tilapia), and Mix (Nile and red tilapia). (**A**) panicle length, (**B**) panicle width, (**C**) panicle weight, (**D**) root length, (**E**) root width, (**F**) 100 seed weight, (**G**) shoot weight, (**H**) number of panicles per five hills, (**I**) number of panicles per hill. Different letters are significantly different according to the LSD test (*p* < 0.05).

**Figure 3 plants-14-01346-f003:**
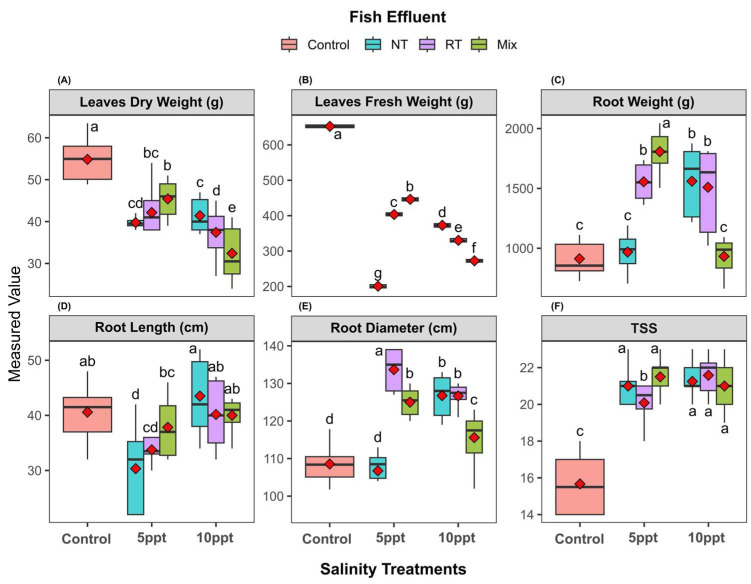
Effects of different treatments on sugar beet traits. Boxplots display the measured values of various traits across three main salinity treatments: 10 ppt, 5 ppt, and control, with seven irrigation effluent sub-treatments: NT (Nile tilapia), RT (red tilapia), and Mix (Nile and red tilapia). (**A**) leaves dry weight, (**B**) leaves fresh weight, (**C**) root weight, (**D**) root length, (**E**) root diameter, (**F**) total soluble solids. Different letters are significantly different according to the LSD test (*p* < 0.05).

**Figure 4 plants-14-01346-f004:**
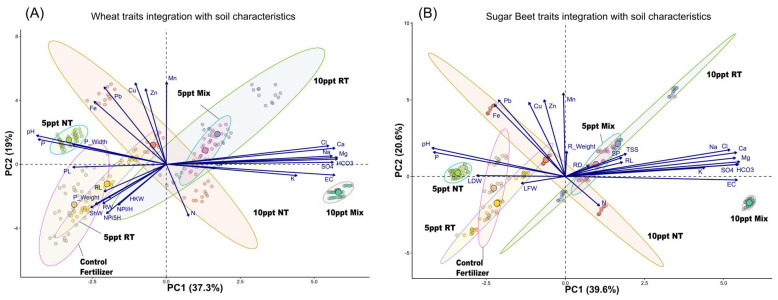
(**A**) Wheat traits integration with soil characteristics: Principal component analysis (PCA) biplot shows the integration of wheat growth and yield with soil characteristics across different treatments. The analysis explains 37.3% of the variance along PC1 and 19% along PC2. PH (plant height), LA (leaf area), SD (stalk diameter), P_Weight (panicle weight), NP/5H (number of panicles per 5 hills), ShW (shoot weight), PL (panicle length), P_Width (panicle width), RW (root width), RL (root length), HKW (100-seed weight), and NPl/H (number of plants per hill). (**B**) Sugar beet traits integration with soil characteristics: Principal component analysis (PCA) biplot shows the integration of sugar beet growth traits with soil characteristics across different treatments. The analysis explains 39.6% of the variance along PC1 and 20.6% along PC2. LDW (leaves’ dry weight), LFW (leaves’ fresh weight), RD (root diameter), R_Weight (root weight), RL (root length), and TSS (total soluble solids).

**Figure 5 plants-14-01346-f005:**
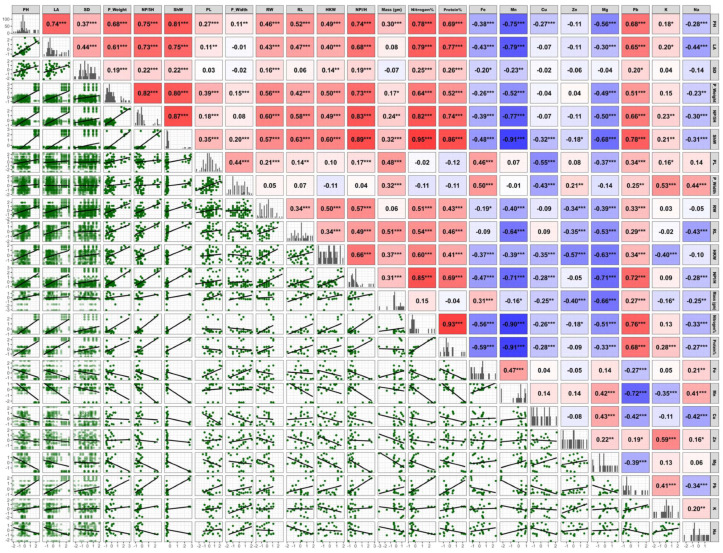
Correlation matrix showing pairwise relationships between wheat growth, yield, and nutrient parameters across different salinity treatments. Color intensity represents the strength and direction of correlations (red = positive and blue = negative). Asterisks indicate statistical significance (* *p* < 0.05, *** p* < 0.01, and *** *p* < 0.001 i.e., significant, highly significant, and very highly significant respectively). Growth and yield traits: PH (plant height), LA (leaf area), SD (stalk diameter), P_Weight (panicle weight), NP/5H (number of panicles per 5 hills), ShW (shoot weight), PL (panicle length), P_Width (panicle width), RW (root width), RL (root length), HKW (100-seed weight), and NPl/H (number of plants per hill). Nutrient analysis: mass (gm), nitrogen (%), protein (%), Fe (iron), Mn (manganese), Cu (copper), Zn (zinc), Mg (magnesium), Pb (lead), K (potassium), and Na (sodium).

**Figure 6 plants-14-01346-f006:**
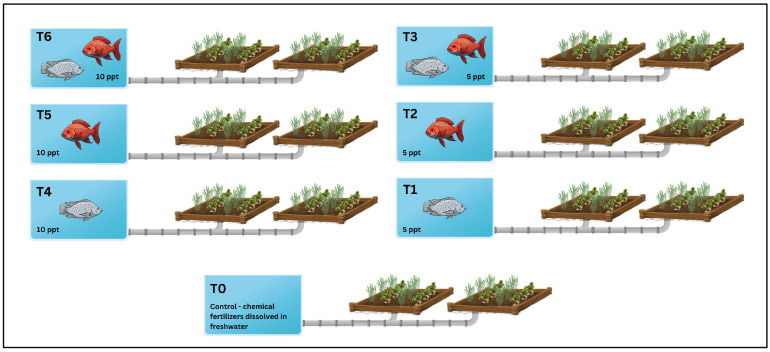
The experiment followed a randomized completely block design of seven irrigation treatments, namely, T0: control—chemical fertilizers dissolved in freshwater, T1: Nile tilapia brackish water fish effluents at 5 ppt, T2: red tilapia brackish water fish effluents at 5 ppt, T3: 1:1 polyculture of Nile and red tilapia under brackish water at 5 ppt, T4: Nile tilapia brackish water fish effluents at 10 ppt, T5: red tilapia brackish water fish effluents at 10 ppt, and T6: 1:1 Polyculture of Nile and red tilapia under brackish water at 10 ppt.

**Table 1 plants-14-01346-t001:** Fish growth performance under different water salinity conditions.

Treatments	Fish	IW ^a^ (g/fish)	FW (g/fish)	WG (g/fish)	FI (g/fish)	FCR	SGR (%/day)	SR%
5 ppt	Nile Tilapia	8.68 ^a^ ± 0.01	160.34 ^e^ ± 0.05	151.66 ^e^ ± 0.04	224.53 ^c^ ± 0.08	1.48 ^a^	1.38 ^c^ ± 0.02	89 ^a^
	Red Tilapia	8.88 ^a^ ± 0.02	161.51 ^c^ ± 0.08	152.63 ^c^ ± 0.09	232.05 ^a^ ± 0.1	1.52 ^a^	1.37 ^c^ ± 0.01	86 ^a^
	Mix	8.52 ^a^ ± 0.012	160.55 ^d^ ± 0.08	152.03 ^d^ ± 0.09	220.93 ^d^ ± 0.05	1.45 ^a^	1.39 ^bc^ ± 0.01	91 ^a^
10 ppt	Nile Tilapia	8.66 ^a^ ± 0.02	144.39 ^f^ ± 0.05	135.73 ^f^ ± 0.06	212.53 ^f^ ± 0.05	1.57 ^a^	1.33 ^d^ ± 0.01	94 ^a^
	Red Tilapia	8.74 ^a^ ± 0.04	174.52 ^a^ ± 0.1	165.78 ^a^ ± 0.06	227.56 ^b^ ± 0.05	1.37 ^a^	1.42 ^a^ ± 0.01	88 ^a^
	Mix	8.76 ^a^ ± 0.03	171.57 ^b^ ± 0.1	162.81 ^b^ ± 0.07	214.63 ^e^ ± 0.1	1.32 ^a^	1.41 ^ab^ ± 0.01	94 ^a^

IW: initial body weight, FW: final body weight, WG: weight gain, FI: feed intake, FCR: feed conversion ratio, SGR: specific growth rate, and SR: survival rate. Means ± SE (*n* = 3) with different letters are significantly different according to the LSD test (*p* < 0.05).

**Table 2 plants-14-01346-t002:** Effects of different irrigation treatments on plant heights of sugar beet at different stages. DAS refers to days after sowing.

Treatment	Fish	30 DAS	45 DAS	75 DAS	105 DAS	135 DAS	165 DAS
Control	Fertilizer	24 ^cd^ ± 0.49	35 ^c^ ± 0.36	47.67 ^a^ ± 0.75	48.79 ^a^ ± 0.86	51.25 ^a^ ± 0.61	53.42 ^a^ ± 0.74
5 ppt	Nile Tilapia	23.04 ^d^ ± 0.49	30.25 ^d^ ± 0.51	35.08 ^c^ ± 0.78	36.18 ^c^ ± 0.79	37.21 ^e^ ± 0.85	37.79 ^d^ ± 0.48
	Red Tilapia	28.5 ^a^ ± 0.75	38.67 ^ab^ ± 1.12	44.5 ^ab^ ± 0.15	48.58 ^a^ ± 0.14	50.6 ^ab^ ± 0.51	51 ^ab^ ± 0.95
	Mix	25.92 ^bc^ ± 0.5	40.42 ^a^ ± 0.54	46.5 ^a^ ± 0.86	47.33 ^a^ ± 0.77	47.75 ^bc^ ± 0.76	49.67 ^b^ ± 0.72
10 ppt	Nile Tilapia	27 ^ab^ ± 0.33	35.63 ^c^ ± 0.44	41.83 ^b^ ± 0.66	45.67 ^ab^ ± 0.81	46.5 ^cd^ ± 0.8	49.54 ^b^ ± 0.83
	Red Tilapia	23.67 ^d^ ± 0.33	35.67 ^c^ ± 0.45	45.17 ^a^ ± 0.86	46.92 ^a^ ± 0.85	47.67 ^bc^ ± 0.45	48.71 ^b^ ± 0.84
	Mix	24.47 ^cd^ ± 0.29	36.92 ^bc^ ± 0.43	41.42 ^b^ ± 0.87	42.88 ^b^ ± 0.8	43.68 ^d^ ± 0.85	45.33 ^c^ ± 0.77

Effects of different irrigation treatments on plant heights of sugar beet at various growth stages (30, 45, 75, 105, 135, and 165 days after sowing). Values represent means ± standard error. Different lowercase letters within the same column indicate significant differences between treatments at each time point (*p* ≤ 0.05). Treatment groups include control (fertilizer only), 5 ppt Nile tilapia effluent, 5 ppt red tilapia effluent, 5 ppt mixed effluent, 10 ppt Nile tilapia effluent, 10 ppt red tilapia effluent, and 10 ppt mixed effluent.

**Table 3 plants-14-01346-t003:** Number of leaves of sugar beet across treatments at different stages. DAS refers to days after sowing.

Treatment	Fish	30 DAS	45 DAS	75 DAS	105 DAS	135 DAS	165 DAS
Control	Fertilizer	6.83 ^b^ ± 0.32	10.33 ^a^ ± 0.48	15.17 ^bc^ ± 0.53	18 ^b^ ± 0.65	20.67 ^bc^ ± 0.51	36.75 ^ab^ ± 0.71
5 ppt	Nile Tilapia	7.17 ^ab^ ± 0.32	9.58 ^a^ ± 0.47	13.92 ^c^ ± 0.54	17.83 ^b^ ± 0.77	19.67 ^c^ ± 0.72	30.67 ^d^ ± 0.28
	Red Tilapia	7.5 ^ab^ ± 0.38	10 ^a^ ± 0.48	16.75 ^ab^ ± 0.65	18.75 ^ab^ ± 0.22	20.83 ^bc^ ± 0.37	32.33 ^d^ ± 0.68
	Mix	8.42 ^a^ ± 0.31	9.5 ^a^ ± 0.47	15.75 ^abc^ ± 0.71	20.33 ^ab^ ± 0.41	23.17 ^ab^ ± 0.53	34.25 ^c^ ± 0.84
10 ppt	Nile Tilapia	7.83 ^ab^ ± 0.27	10 ^a^ ± 0.28	16.5 ^ab^ ± 0.51	21.25 ^a^ ± 0.71	24.58 ^a^ ± 0.82	38.33 ^a^ ± 0.59
	Red Tilapia	7.75 ^ab^ ± 0.3	10.5 ^a^ ± 0.54	17.67 ^a^ ± 0.58	19.17 ^ab^ ± 0.63	21.92 ^abc^ ± 0.61	38.58 ^a^ ± 0.66
	Mix	8.33 ^a^ ± 0.36	10.33 ^a^ ± 0.33	15.75 ^abc^ ± 0.51	20.67 ^a^ ± 0.76	21.67 ^bc^ ± 0.73	36.17 ^b^ ± 0.8

Effects of different irrigation treatments on the number of leaves in sugar beet at various growth stages (30, 45, 75, 105, 135, and 165 days after sowing). Values represent means ± standard error. Different lowercase letters within the same column indicate significant differences between treatments at each time point (*p* ≤ 0.05). Treatment groups include control (fertilizer only), 5 ppt Nile tilapia effluent, 5 ppt red tilapia effluent, 5 ppt mixed effluent, 10 ppt Nile tilapia effluent, 10 ppt red tilapia effluent, and 10 ppt mixed effluent.

**Table 4 plants-14-01346-t004:** The climatic data during the experimental season.

Growing Season	Air Temp. (°C)	Relative Humidity (%)	Short Wave Solar Radiation (W/m^2^)	Wind Speed (m/s)	Total Sun Hours/Day	ETo
**Sep-23**	27.94	68.73	253.75	1.59	11.38	9.55
**Oct-23**	23.83	77.37	183.16	1.45	10.46	6.22
**Nov-23**	20.65	74.21	144.76	1.45	9.61	4.64
**Dec-23**	16.45	74.03	118.61	1.40	9.13	3.27
**Jan-24**	14.20	69.08	132.60	1.17	9.34	3.47
**Feb-24**	14.00	74.14	170.13	1.53	10.07	4.59
**Mar-24**	17.40	64.88	221.54	1.56	10.99	6.70
**Apr-24**	22.24	65.32	274.04	1.77	11.88	9.39

Monthly mean values of climatic parameters during the experimental period (September 2023 to April 2024). ETo represents reference evapotranspiration.

## Data Availability

The original contributions presented in this study are included in the article. Further inquiries can be directed to the corresponding author.

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
