# Peer review of "Evaluating the Growth Performance of Nile and Red Tilapia and Its Influence on Morphological Growth and Yield of Intercropped Wheat and Sugar Beet Under a Biosaline Integrated Aquaculture–Agriculture System"

_plants, 2025, doi:10.3390/plants14091346_

Round 1

Reviewer 1 Report

Comments and Suggestions for Authors

This is very good idea study for sustainable agriculture but few points need to address before move further process.
1- The abstract of this study need to re-written and add few results along with effects of treatment then compared results and written conclusion.
2- The introduction part need to revised and add more details of Integrated Aquaculture-Agriculture (IAA) systems instead of salinity impact on crop growth. furthermore, link how IAA systems helps under saline conditions. does these techniques can be used for higher level?
3- Table 1 all values have same Standred error that is surprising, that need to check, figure 1 and 2 font need to improved and all results section need to revised .because author said it is intercropping experiment but sugar beet growth monitor till 165 DAS and wheat only 90 DAS. this should need to explain because it is confusing. all figures need to reside because font is low.
4- Discussion section need to revised link your finding with pervious finding, explain the cost benefits of this study. does this techniques poor farmer will adopt ? does this study give some direction for policy makers to use this techniques or not?
5- material and method section should add fish water used for irrigation after how many days growth of fish in it.
6- conclusion need to re-written 

Reviewer 2 Report

Comments and Suggestions for Authors

The article provides a detailed study on the growth performance of Nile tilapia and red tilapia under different salinity conditions,as well as the impact of their aquaculture wastewater on the growth and yield of wheat and sugar beet.The innovation of the study is indeed commendable.Below are some suggestions for improvement,and in the revision response,I would like to see a detailed reply letter(for example,the specific locations of revisions should be pointed out in the reply letter):

  • Line 62-65,I would like to see the percentage decrease in the yield of wheat and sugar beet under different salinity conditions,expressed as a percentage(%).
  • The introduction could further emphasize the innovation of the study.For instance,why Nile tilapia and red tilapia were chosen,and why wheat and sugar beet were selected as research objects,especially given the significant decrease in wheat yield.
  • Line 80-85,I would like to cite“https://doi.org/10.1016/j.fcr.2025.109747”to further highlight the hazards of salt stress.
  • In the methods section,the specific experimental procedures could be described in more detail,such as how to prepare aquaculture wastewater and how to conduct irrigation.Since this article can serve as a reference for others'experiments,it would be helpful for future detailed studies.
  • In the discussion section,the specific physiological and biochemical impact mechanisms of different salinities on fish and crops could be explored more deeply.For example,why red tilapia performs better under higher salinity,while wheat performs poorly under saline conditions.
  • In the discussion section,the specific impact of the nutritional components of aquaculture wastewater on crop growth could be further discussed,as well as how to optimize the use of wastewater to increase crop yield.
  • Consider adding some supplementary figures,such as growth curves of fish under different salinities,to better illustrate dynamic changes.
  • The conclusion section could more specifically point out the practical application value of the study,such as how to promote IAAS in saline-alkali areas.
  • Future research directions could be proposed,such as how to further optimize the IAAS system to improve its efficiency and sustainability.

In summary,the article indeed provides a research basis for future saline water irrigation.After the above revisions,it could be considered for acceptance.

Reviewer 3 Report

Comments and Suggestions for Authors

 The main comment concerns the structure of the article. The "Materials and Methods" chapter should be at the beginning of the article, after the introductory chapter. It describes the research sites, research design, and methods used. This is in connection with the established research hypothesis.

Reviewer 4 Report

Comments and Suggestions for Authors

Dear Authors, you should address my comments highlighted across the text.

Comments on the Quality of English Language

Moderate editing is needed.

Round 2

Reviewer 1 Report

Comments and Suggestions for Authors

accept in current form

Author Response

Thanks for your efforts towards our manuscript

Reviewer 2 Report

Comments and Suggestions for Authors

please added the references in line 112-122

Author Response

Thanks

The references have been added in the revised manuscript.

Reviewer 4 Report

Comments and Suggestions for Authors

Dear Authors, the manuscript can be accepted in present form for publication in Plants, in my opinion. 

Author Response

Thanks for your efforts and time